

# Physical activity, problematic smartphone use, and burnout among Chinese college students

Lianghao Zhu[1,*], Junli Hou[1,*], Bojun Zhou[2], Xi Xiao[1], Jingqiang Wang[1] and Wanping Jia[3]

[1] School of Physical Education, Hubei Business College, Wuhan, China
[2] School of Kinesiology, Beijing Sport University, Beijing, China
[3] Center for International Education, Philippine Christian University, Manila, Philippines
* These authors contributed equally to this work.

## ABSTRACT

The aim of this study was to investigate the association between physical activity (PA), problematic smartphone use (PSU), and burnout, as well as to identify whether there is a mediating role for PSU. We recruited 823 college students ($M_{age}$ = 18.55, $SD$ = 0.83) from Wuhan, China, in December 2022, including 499 males and 324 females. Demographic information, the International Physical Activity Questionnaire-Short Form (IPAQ-SF), the Smartphone Addiction Scale-Short Version (SAS-SV), and the Maslach Burnout Inventory-Student Survey (MBI-SS) were used for assessments. Pearson correlation analysis showed that PA was significantly associated with PSU ($r$ = −0.151, $p$ < 0.001), PSU was significantly associated with burnout ($r$ = 0.421, $p$ < 0.001), and the association between PA and burnout was not statistically significant ($r$ = −0.046, $p$ > 0.05). The results of the mediation model test showed that PA could not predict burnout directly; it instead predicted burnout entirely indirectly through PSU. Furthermore, PSU mediated the predictive effect of PA on exhaustion and cynicism. In conclusion, there is no direct connection between PA levels and burnout. PA indirectly affects burnout through PSU, but does not fully apply to the three different dimensions of exhaustion, cynicism, and professional efficacy.

## INTRODUCTION

Burnout was described as the experience of exhaustion in which individuals become cynical about the worth of their occupation and doubt their capacity to function (*Maslach, Jackson & Leiter, 1996*). Physical and behavioral signs of burnout, according to the definition of *Maslach, Jackson & Leiter (1996)*, involve three dimensions: exhaustion, cynicism, and lack of professional efficacy. Although students are neither officially employed nor hold employment, it is possible that burnout also affects them. However, their primary activities can be seen as "work" from a psychological standpoint (*Jacobs & Dodd, 2003*; *Ye, Huang & Liu, 2021*). The admittance to high schools and universities, and

Corresponding author
Wanping Jia, zzz430079@126.com

consequently pupils' job prospects, is determined by extremely competitive exams used in the rigorous Chinese educational system. At the same time, the quality of teaching is assessed by students' examination results, so teachers put enormous pressure on students (*Hu & Schaufeli, 2009*). A cross-sectional study of 22,983 Chinese university students showed that more than half of the students suffered from academic burnout: mild burnout, severe burnout, and very severe burnout accounted for 55.16%, 3.55%, and 1.28%, respectively (*Liu et al., 2023*). Previous studies showed that higher levels of burnout are associated with lower academic achievement (*Madigan & Curran, 2021*) and an increased risk of withdrawal intentions (*Williams, Dziurawiec & Heritage, 2018*), emotional dysregulation, perpetration (*Cooper et al., 2017*), sleep disorders (*Pagnin et al., 2014*), depression (*Peterka-Bonetta et al., 2019*), and suicide (*Bitran et al., 2019*). Simultaneously, the quality of work, professionalism, ethical behavior, and empathy were negatively impacted (*Bitran et al., 2019*). Therefore, given the many potential hazards of burnout on physical and mental health as well as daily routines, the main triggers of burnout in college students need more consideration in order to alleviate it.

As of June 2022, the scale of China's mobile phone Internet users was 1.047 billion, the user scale of short videos reached 962 million, and the user scale of instant messaging reached 1.027 billion. The tendency of these three sorts of users tending to be more youthful is particularly pronounced (*China Internet Network Information Center, 2023*). While the popularity of smartphones has fueled the development of productivity in society, overuse of smartphones can also have some negative effects (*Wu et al., 2021*). Researchers from mainland China found that problematic smartphone use (PSU) may be an antecedent to burnout in school settings, which provides some insight into future interventions to deal with burnout in college students (*Zhang et al., 2021a*, *2021b*; *Meng et al., 2023*). On the other hand, physical activity (PA) has the advantages of being accessible, low-cost, and effective in alleviating burnout, driving researchers to explore it (*Li et al., 2021*; *Lee et al., 2020*). Concurrently, an increasing number of people have begun to pay attention to PA as a protective effect against PSU (*Xiao et al., 2022*; *Yang et al., 2021*). To our knowledge, few studies have combined PA with PSU in dealing with burnout-related issues, although this approach has been applied in observational studies of irrational procrastination (*Shi et al., 2021*), sleep quality (*Gao et al., 2023*), suicidality (*Xie et al., 2020*), and depression (*Xie et al., 2019*). In response to the aforementioned needs of the cultural context and the limitations of the available evidence, the purpose of this study was to explore the association between PA, PSU, and burnout, as well as the mediating role of PSU, providing an empirical profile from a sample of Chinese college students.

## Physical activity as a predictor of burnout

Physical activity is any bodily movement produced by skeletal muscles that requires energy expenditure (*Bull et al., 2020*). *Naczenski et al. (2017)* and *Dreher, Dößereck & Lachtermann (2020)* synthesized evidence regarding the relationship between PA and burnout, with PA negatively associated with burnout in longitudinal studies and somewhat mitigating the threat of burnout in intervention studies. In particular, quantification of this evidence by the standardized index of convergence (SIC; *Wielenga-Meijer et al., 2010*)

provided moderately strong evidence for a negative relationship. Despite the lack of high quality studies, PA was considered to be an effective medium for reducing burnout.

Regarding psychological operating mechanisms, regular PA could reduce the risk of chronic stress by mentally disengaging from work during leisure time (*Sonnentag, 2012*). For people who perform sedentary jobs, strenuous leisure time PA once or twice a week was associated with a low risk of future exhaustion (*Bernaards et al., 2006*). *De Vries et al. (2016)* found that an increase in moderate intensity physical activity (MPA) was associated with a reduction in fatigue related to work. The changes in PA were negatively correlated with changes in physical fatigue, emotional exhaustion, and cognitive weariness (*Lindwall et al., 2014*), and individuals who performed PA were less likely to be diagnosed with burnout (*Jonsdottir et al., 2010*). Consequently, PA can be effective against the key component of burnout (*i.e.*, exhaustion).

According to *Ahola et al. (2012)*, insufficient PA was linked not only to the exhaustion dimension but also to the cynicism dimension and worse occupational efficacy. The reason may have been the result of a long-term mismatch between work demands and available resources (*Bakker & Demerouti, 2007*; *Schaufeli & Bakker, 2004*), in other words, the failure to meet career expectations. Cynicism, also known as depersonalization, is characterized by an indifferent or detached attitude toward work in general and the people with whom one works, resulting in a lack of interest in work and a sense that work has lost its significance (*Koutsimani, Montgomery & Georganta, 2019*; *Maslach, Jackson & Leiter, 1996*). Conceptually, cynicism and dedication (one of the subdimensions of work engagement) are the opposites of each other (*Schaufeli & Bakker, 2004*). *Kara, Gürbüz & Öncü (2019)* discovered that physically active individuals showed greater commitment to the importance, willingness, pride, and challenge of their work.

Furthermore, regular participation in PA can enhance people's self-efficacy, which can even "spill over" into work areas, resulting in greater self-approval. In accordance with the effort-recovery model (*Meijman & Mulder, 1998*), leisure-time physical activities (even if they require energy expenditure) afford an opportunity to replenish stress resources and acquire new ones. For example, PA outside of work could provide a recovery through physical self-perception and self-esteem to counteract the harmful future effects of stress and fatigue at work on physical and mental health (*Feuerhahn, Sonnentag & Woll, 2014*; *Rook & Zijlstra, 2006*).

In summary, the available evidence supports the negative association between PA and burnout, but the participants were mainly employees of the health and social security systems or medical students, and it is impossible to determine whether the results of previous studies are heterogeneous between college students of different majors. Thus, a hypothesis is proposed. Hypothesis 1: There will be a negative correlation between PA and burnout.

## Problematic smartphone use as a predictor of burnout

Problematic smartphone use is broadly described as a recurring desire to use a smartphone in an uncontrollable manner that alters daily functioning (*Busch & McCarthy, 2021*; *Horwood & Anglim, 2018*). This uncontrolled usage manifests itself in two ways: a

maladaptive dependency (*Chen et al., 2017*) and a proclivity to use the smartphone without being removed from it (*Cho & Lee, 2017*). *Busch & McCarthy (2021)* proposed that emotional health issues (*e.g.*, anger, stress, anxiety, depression, loneliness, self-esteem, and happiness) are the most typical consequence of PSU and that adolescents are especially at risk for them. Similarly, it is vital to comprehend how PSU, as an antecedent, might lead to burnout.

A large body of research has shown that PSU is positively related to burnout in school contexts (*Hao et al., 2021*, *2022*; *Li et al., 2021*; *Zhang et al., 2021a*, *2021b*). First, according to resource depletion theory, individuals have finite mental resources for information processing, and the preservation of attention is dependent on the coupling of those available resources. In particular, cognitive resources deplete faster than they can be replenished (*i.e.*, cognitive load accumulation) as the duration of the task increases, resulting in difficulty concentrating (*Head & Helton, 2014*; *Helton & Russell, 2011*). PSU occupies a significant amount of the time and cognitive resources of adolescents, which should be used for thoughtful consideration of relevant curriculum and academics in the school setting. Adolescents who use their smartphones excessively are likely to fall behind in their studies, creating feelings of stress (*Zhang et al., 2021a*).

Second, smartphone usage could be divided into two types based on process and content gratification (*Song et al., 2004*). "Process usage" focuses on recreational functions such as news consumption, entertainment, and leisure (*Hao et al., 2022*; *Van Deursen et al., 2015*). Since pleasant experiences act as a reward, the likelihood of habitual use of smartphones increases (*Yang & Tung, 2007*), while personal interest in learning and working diminishes. This is a manifestation of entertainment or escapism (*Wang et al., 2015*). "Social usage" describes the use of social communication channels such as Facebook, text messaging, and phone calls (*Hao et al., 2022*; *Van Deursen et al., 2015*). *Walburg, Mialhes & Moncla (2016)* discovered that levels of school-related burnout were strongly associated with a higher frequency of Facebook use.

Third, there was sufficient evidence that PSU is negatively associated with academic performance (*Bai, Chen & Han, 2020*; *Grant, Lust & Chamberlain, 2019*; *Hawi & Samaha, 2016*; *Lepp, Barkley & Karpinski, 2015*; *Samaha & Hawi, 2016*; *Troll, Friese & Loschelder, 2021*). Excessive smartphone use resulting in a decline in academic performance (*Grant, Lust & Chamberlain, 2019*) is a negative feedback message for students, as it opposes the desired goals originally intended to be achieved and may even affect subsequent professional efficacy (*Adams et al., 2020*). In summary, PSU is a key cause of burnout. Hypothesis 2: There will be a positive correlation between PSU and burnout.

## Association between PA and PSU

In the past, several studies have noted that high-risk smartphone users tend to exhibit less PA (*Kim, Kim & Jee, 2015*; *Pereira et al., 2020*). The explanation for this was that sedentary behavior could encompass many usage situations for smartphone functions (*Mansoubi et al., 2014*; *Rosenberg et al., 2010*) and that PSU had a positive correlation with sedentary behavior (*Barkley, Lepp & Salehi-Esfahani, 2016*; *Barkley & Lepp, 2016*; *Xiang et al., 2020*). High-frequency users were more likely to forego opportunities for PA in favor of using

their smartphones (*Lepp et al., 2013*), and even the intensity of the exercise they were performing would be disturbed (*Barkley & Lepp, 2016*), compared to low-frequency users. The above investigations on the connection between PSU and PA virtually invariably support the thesis that PA levels can be adversely affected by PSU.

In fact, there was a bidirectional relationship between PSU and PA, with moderate to vigorous physical activity (MVPA) effective in preventing smartphone overuse (*Kim & Lee, 2022*). A study on 24-h movement behavior conducted by *Ren et al. (2022)* demonstrated that the more time spent on the MVPA, the less the PSU phenomenon occurred. On the other hand, reduction of individual inhibitory control caused a decrease in motivational seeking and suppression of desire, which resulted in excessive addictive behavior, according to the Interaction of Person-Affect-Cognition-Execution (I-PACE) model for addictive behaviors (*Brand et al., 2016*, *2019*). Exercise was one way to enhance inhibitory control (*Fan et al., 2021*), and *Zeng et al. (2022)* and *Zhang et al. (2022)* found that exercise could indirectly affect PSU through self-control. It was practicable to move away from the immediate hedonism of PSU by using capacity-enhancing strategies targeted at improving individuals' self-discipline and rational management (*Busch & McCarthy, 2021*). Individuals have limited willpower to maintain self-regulation and can benefit from encouraging smartphone users to engage in more physical activities such as sports, outdoor activities, dance, and yoga (*Mahapatra, 2019*). Thus, a hypothesis is proposed: Hypothesis 3: There will be a negative correlation between PA and PSU.

## PSU as a mediator in the association between PA and burnout

In a literature review of the antecedents of PSU, deficits in personal self-regulation and distraction from negative emotions provide important explanations for our understanding of the emergence of PSU (*Busch & McCarthy, 2021*). Self-control is the ability to resist temptations, break habits, and override urges (*Hagger et al., 2019*). *Zou et al. (2016)* found that aerobic exercise could be a potentially effective intervention to enhance self-control in individuals. According to the self-control strength model (*Baumeister et al., 1998*), individuals who take actions that call for self-control resources temporarily spend certain finite resources that are not instantly regenerated. The recovery of self-control can surpass the ceiling of the initial self-control resources after restricted resources have been exhausted when both PA and PSU are perceived as stable behaviors over time. Self-control as an individual trait increases as a result of this process, which lessens reliance on smartphones (*Zhong, Wang & Zhang, 2021*). In addition, PA and exercise can be used to treat depression, anxiety, and panic disorders in a process that involves complex neurobiological mechanisms (*Ströhle, 2009*). We believe that physical activity engagement may undermine one of the triggers for PSU, and that this trigger is motivated by the avoidance of negative emotions.

As far as the consequences of the PSU are concerned, it may affect the professional performance of the user (*Busch & McCarthy, 2021*). When high-risk smartphone users are without access to their smartphones, they become impatient, and the resulting tendency towards boredom is exacerbated (*Ruiz-Palmero et al., 2019*). It further makes them uninterested and even unenthusiastic about learning. For individuals with higher levels of

boredom proneness, they may pause, delay, or terminate necessary tasks they are working on (*e.g.*, school-related tasks) after experiencing interruptions from smartphone interruption notifications (*Elhai et al., 2021*). The suspension of such tasks is a drag because of the need to convert costs into productivity when trying to resume (*Salvucci & Taatgen, 2008*). It may bring forward the psychological reaction of exhaustion in the course of their work engagement. Furthermore, there is evidence that PSU influences students' achievement motivation and academic performance (*Arefin et al., 2018*), which may lead to an attenuation of professional efficacy. In conclusion, the antecedents and consequences of PSU can provide a novel explanatory mechanism for the association between PA and burnout (including its three dimensions). To be further verified, the following hypothesis is proposed: Hypothesis 4: PSU mediates the relationship between PA and burnout, which would extend to exhaustion, cynicism, and professional efficacy.

## MATERIALS AND METHODS

### Research design

Using the stratified cluster random sampling technique, we distinguished eight mutually independent strata based on differences in faculties, and then natural classes were used as clusters. A sample of three classes from each of the eight different faculties was randomly taken, and a total of 24 classes were invited to participate in the survey. This study employed a cross-sectional design in which participant demographic information, IPAQ-SF, SAS-SV, and MBI-SS measures were collected at a single point in time. With PA as the independent variable, PSU as the mediator, and burnout and its three dimensions as the dependent variables, we examined the intrinsic associations between the variables.

### Participants

The participants were 823 college students (60.6% male) ranging from 16 to 21 years old, with a mean age of 18.55 years old and a standard deviation of 0.83. They were retained from a sample of 866 college students due to the fact that 43 participants did not successfully submit their data, as well as the fact that the submitted answers were invalid. In the Chinese educational system, the training length for undergraduate education is 4 to 5 years and 3 years for vocational education. The majority of participants (78.7%) were receiving vocational education. In terms of family situation, 69.7% of these students are not only children; 55.9% live in the countryside; and 74.1% have an annual household income between RMB 10,000 and 150,000.

### Procedures

The study was approved by the Ethics Committee of Hubei University of Medicine (NO: 2022-TH-068) and was conducted at a college in Wuhan, Central China. In December 2022, with the opportunity for an all-school faculty meeting, we provided intensive training for teachers involved in data collection during the final phase of the meeting. During the first 10 min of regular class, each participant was notified about the purpose of the study, and written informed consent was provided. Then, while using multimedia equipment to display the QR code of the online questionnaire on the screen, the teacher

told the students to read the instructions of the questionnaire carefully and start answering. Students completed the questionnaire online by scanning the code on their own smartphones. They submitted their answers anonymously and received no compensation at the end of the study.

## Measures

### Demographic information

Gender, age, educational qualification, type of household residence, annual family income, and "only child or not" were among the demographic data sought.

### International physical activity questionnaire-short form (IPAQ-SF)

PA was measured with the Chinese version of the IPAQ-SF (*Macfarlane et al., 2007*). The questionnaire was self-reported with reference to PA recollections of the past 7 days. The IPAQ-SF inquired about three specific types of activity that were performed in the four domains (*i.e.*, leisure-time physical activity, domestic and gardening activities, work-related physical activity, and transport-related physical activity). Walking, moderate-intensity activity (MPA), and vigorous-intensity activity (VPA) were the specific types of activities evaluated. According to guidelines for data processing and analysis of the IPAQ, we used MET-minutes/week as a measure of the volume of activity. Total physical activity MET-minutes/week = sum of Walking + Moderate + Vigorous MET-minutes/week scores. The sitting item was excluded from the PA summary score since it represents a supplementary indicator variable of sedentary behavior. For more details on IPAQ scoring see the Supplemental Material. Furthermore, with reference to the three levels of physical activity proposed in the Supplemental Material, a combination of various types of physical activity of at least 600 MET minutes/week (category 2) for the medium level and at least 3000 MET minutes/week (category 3) for the high level. The PARS-3 actually catches more leisure-time physical activity, whereas the IPAQ captures a wider range.

### Smartphone addiction scale-short version (SAS-SV)

PSU was measured with the 10-item SAS-SV (*Kwon et al., 2013*), whose content includes health and social disorders, withdrawal, and tolerance related to smartphone use. It is important to note that in the eighth item, "Twitter or Facebook" was replaced with the more conceptual "social networking platforms" because Twitter and Facebook were not the mainstream tools for Chinese smartphone users. Each item is presented on a 6-point Likert scale ranging from "1 = Strongly disagree" to "6 = Strongly agree." The cut-off values of 31 (males) and 33 (females) were derived from ROC analyses. In our sample, the confirmatory factor analysis (CFA) results, after establishing residual correlations by modification indices, were as follows: CFI = 0.946, GFI = 0.931, RMSEA = 0.098, and SRMR = 0.057, indicating an acceptable model fit. The Cronbach alpha was 0.908.

### Maslach burnout inventory-student survey (MBI-SS)

Burnout was measured with the 15-item MBI-SS (*Schaufeli et al., 2002*) in three dimensions: exhaustion (EX), cynicism (CY), and professional efficacy (PE). Its original version was the Maslach Burnout Inventory-General Survey (*Schaufeli et al., 1996*), for

which we obtained a license to use from the publisher, Mind Garden. *Hu & Schaufeli*'s *(2009)* study showed that the scale was applicable to Chinese students. All items are scored on a 7-point frequency rating scale ranging from "0 = Never" to "6 = Always." High scores on EX and CY and low scores in PE are indicative of burnout (*i.e.*, all items of professional efficacy are reverse scored). The EX, CY, and rPE subscale scores had clinically verified cut-off values of 12.5, 7.5, and 10.5 for each, respectively (*Wickramasinghe, Dissanayake & Abeywardena, 2018*). After establishing residual correlations by modification indices, the results of the second-order validation factor analysis were as follows: CFI = 0.951, GFI = 0.911, RMSEA = 0.083, and SRMR = 0.052, indicating an acceptable model fit. The Cronbach alpha for the entire scale in our sample was 0.837.

## Statistical approach

Means and standard deviations for continuous variables were calculated by SPSS 26.0, as were frequencies and percentages for categorical variables. First, independent sample *t*-tests and one-way ANOVA were used to explore whether the sample of various social characteristics differed in the scores reported on the PA, PSU, and burnout. Meanwhile, we supplemented the corresponding effect sizes. According to *Ferguson (2009)*, the small, medium, and large effect sizes for Cohen's *d* were 0.41, 1.15, and 2.70, respectively, while for $\eta^2$, they were 0.04, 0.25, and 0.64. Subsequently, Pearson correlation analysis was used to explore the correlation between the variables. Finally, using the PROCESS macro in *Hayes (2013)* SPSS 26.0, "Model 4" was selected to test whether PSU plays a mediating role in the relationship between PA and burnout when controlling for gender. As suggested by *MacKinnon, Krull & Lockwood (2000)* and *Zhao, Lynch & Chen (2010)*, the test procedure for indirect effects was continued even if the total effect was not significant. The test level α was 0.05.

# RESULTS

## Descriptive statistics

Scores were reported for the three main instruments IPAQ-SF, SAS-SV, and MBI-SS in Table 1. We found that 318 (38.6%) students reached the moderate level and 379 (46.1%) students reached the high level. A total of 241 (48.3%) male and 134 (41.4%) female students had problems with smartphone use, respectively. The reality was that there was exhaustion in 253 (30.7%) students, cynicism in 380 (46.2%) students, and a lack of professional efficacy in 633 (76.9%) students.

## Independent samples *t*-tests and one-way ANOVA

Table 2 showed the results of the independent sample *t*-test, with male students scoring significantly higher on PA ($p < 0.001$, Cohen's $d = 0.436$) and burnout ($p < 0.05$, Cohen's $d = 0.175$) than female students, so we subsequently brought gender into the regression analysis as a control variable. There were no significant differences in scores on PA, PSU, or burnout between students with various levels of education, types of residence, or "only child or not" (all $p > 0.05$, all Cohen's $d < 0.41$). In addition, the results of the one-way ANOVA revealed no significant differences in PA ($p > 0.05$, $\eta^2 = 0.004$), PSU ($p > 0.05$,

**Table 1 Descriptive statistics for all continuous and categorical variables.**

| Variables | n (%) or M ± SD |
|---|---|
| Age | 18.55 ± 0.83 |
| *Gender* | |
| Male | 499 (60.6) |
| Female | 324 (39.4) |
| *Education level* | |
| Vocational education | 648 (78.7) |
| Undergraduate education | 175 (21.3) |
| *Only child or not* | |
| Yes | 249 (30.3) |
| No | 574 (69.7) |
| *Type of residence* | |
| Countryside | 460 (55.9) |
| City or town | 363 (44.1) |
| *Annual family income* | |
| RMB 10,000 to 150,000 | 610 (74.1) |
| RMB 150,001 to 300,000 | 147 (17.9) |
| RMB 300,001 or more | 66 (8.0) |
| *IPAQ-SF* | |
| Vigorous MET-minutes/week | 1,618.030 ± 2,302.092 |
| Moderate MET-minutes/week | 834.280 ± 1,097.699 |
| Walking MET-minutes/week | 1,389.500 ± 1,709.041 |
| PA | 3,841.810 ± 3,907.981 |
| *SAS-SV* | |
| PSU | 30.170 ± 9.605 |
| *MBI-SS* | |
| EX | 10.110 ± 5.742 |
| CY | 6.940 ± 4.773 |
| PE | 20.860 ± 6.957 |
| rPE | 15.140 ± 6.957 |

Notes:
Vigorous MET-minutes/week = 8.0 * vigorous-intensity activity minutes * vigorous-intensity days; Moderate MET-minutes/week = 4.0 * moderate-intensity activity minutes * moderate-intensity days; walking MET-minutes/week = 3.3 * walking minutes * walking days; PA (Total physical activity MET-minutes/week) = sum of walking + moderate + vigorous MET-minutes/week scores.

$\eta^2 = 0.005$), or burnout ($p > 0.05$, $\eta^2 = 0.004$) scores between students from the three levels of annual household income.

## Correlation analysis, multiple linear regression analysis, and tests for mediating effects

The results of the correlation analysis in Table 3 showed that PA was significantly negatively associated with PSU ($r = -0.151$, $p < 0.001$), PSU was significantly positively associated with burnout ($r = 0.421$, $p < 0.001$), while the negative association between PA and burnout was not statistically significant ($r = -0.046$, $p > 0.05$). Similarly, PA did not

**Table 2 Differences in the mean of scores on PA, PSU and burnout for samples with various characteristics.**

| | Categories | PA | $t$ or $F$ | Cohen's $d$ or $\eta2$ | PSU | $t$ or $F$ | Cohen's $d$ or $\eta2$ | Burnout | $t$ or $F$ | Cohen's $d$ or $\eta2$ |
|---|---|---|---|---|---|---|---|---|---|---|
| Gender | Male | 4,480.450 ± 4,193.067 | 6.287*** | 0.436 | 30.300 ± 9.757 | 0.473 | 0.034 | 33.060 ± 12.763 | 2.445* | 0.175 |
| | Female | 2,858.220 ± 3,187.204 | | | 29.970 ± 9.378 | | | 30.860 ± 12.386 | | |
| Education level | Vocational education | 3,845.660 ± 3,932.313 | 0.054 | 0.005 | 30.170 ± 9.490 | −0.004 | 0.000 | 32.320 ± 12.485 | 0.560 | 0.047 |
| | Undergraduate education | 3,827.540 ± 3,827.593 | | | 30.170 ± 10.050 | | | 31.720 ± 13.289 | | |
| Only child or not | Yes | 3,950.130 ± 4,034.654 | 0.524 | 0.039 | 30.390 ± 10.067 | 0.434 | 0.033 | 32.220 ± 13.188 | 0.038 | 0.003 |
| | No | 3,794.810 ± 3,854.374 | | | 30.070 ± 9.405 | | | 32.180 ± 12.427 | | |
| Type of residence | Countryside | 3,923.520 ± 4,031.380 | 0.675 | 0.048 | 29.840 ± 8.844 | −1.087 | −0.077 | 32.590 ± 12.186 | 1.004 | 0.070 |
| | City or town | 3,738.260 ± 3,748.738 | | | 30.590 ± 10.489 | | | 31.700 ± 13.223 | | |
| Annual family income | RMB 10,000 to 150,000 | 3,708.990 ± 3,828.226 | 1.653 | 0.004 | 30.140 ± 9.297 | 1.946 | 0.005 | 32.360 ± 12.177 | 1.448 | 0.004 |
| | RMB 150,001 to 300,000 | 4,085.750 ± 3,614.016 | | | 31.120 ± 9.854 | | | 32.650 ± 13.263 | | |
| | RMB 300,001 or more | 4,525.990 ± 5,070.924 | | | 28.320 ± 11.548 | | | 29.680 ± 15.280 | | |

Notes:
* $p < 0.05$.
*** $p < 0.001$.
Burnout = EX + CY + rPE.

**Table 3 Results of Pearson correlation analysis.**

| | $M$ | $SD$ | PA | PSU | Burnout | EX | CY | PE |
|---|---|---|---|---|---|---|---|---|
| PA | 3,841.810 | 3,907.981 | 1 | | | | | |
| PSU | 30.170 | 9.605 | −0.151*** | 1 | | | | |
| Burnout | 32.200 | 12.654 | −0.046 | 0.421*** | 1 | | | |
| EX | 10.110 | 5.742 | −0.061 | 0.526*** | 0.770*** | 1 | | |
| CY | 6.940 | 4.773 | −0.079* | 0.451*** | 0.797*** | 0.744*** | 1 | |
| PE | 20.860 | 6.957 | −0.020 | −0.023 | −0.636*** | −0.065 | −0.151*** | 1 |

Notes:
* $p < 0.05$.
*** $p < 0.001$.

have statistically significant associations with EX and PE (all $p > 0.05$) on any of the three dimensions of burnout. PSU was positively associated with EX ($r = 0.526$, $p < 0.001$) and CY ($r = 0.451$, $p < 0.001$).

"Model 4" of the PROCESS macro in SPSS 26.0 was used to examine the relationships between the variables, with gender as the control variable, PA as the independent variable, PSU as the mediator, and burnout, EX, CY, and PE as the dependent variables, respectively. The results of multiple linear regression analysis in Fig. 1 found that PA had a
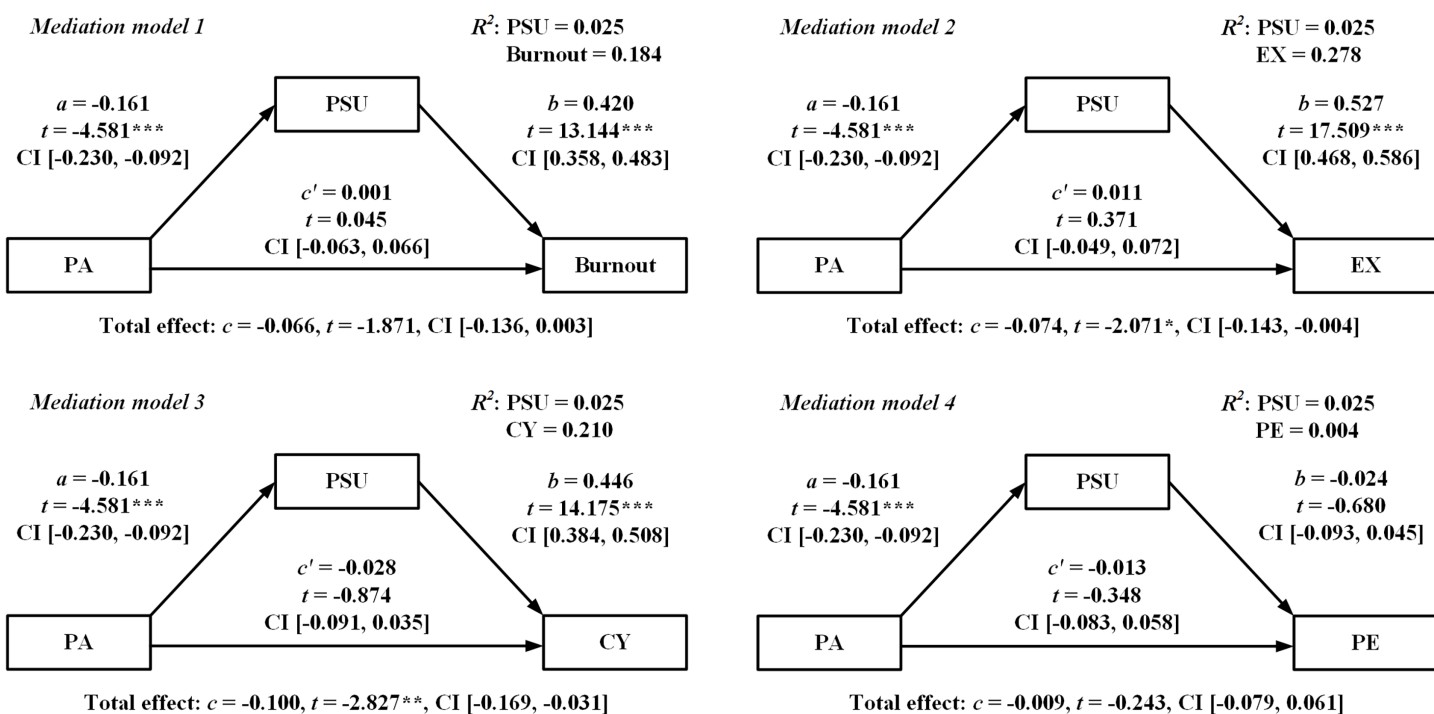

**Figure 1 Results of multiple linear regression analysis.** *Notes:* *p < 0.05, **p < 0.01, ***p < 0.001. Regression coefficients were standardized. The parameters of the control variable (*i.e.*, gender) were not shown in the model. Level of confidence for all confidence intervals in output: 95%.

significant predictive effect on PSU ($\beta = -0.161$, $p < 0.001$, 95% CI [$-0.230$ to $-0.092$]), but no statistically significant direct effects on burnout, EX, CY, or PE (all $p > 0.05$, 0 was contained within LLCI and ULCI). PSU had a significant predictive effect on burnout, EX, and CY with $\beta$ of 0.420, 0.527, and 0.446, respectively, all with $p$-values less than 0.001.

With respect to gender (male = 1, female = 0), we found that it did not significantly predict PSU ($\beta = 0.101$, $p > 0.05$, 95% CI [$-0.041$ to 0.242]). Gender significantly predicted burnout ($\beta = 0.159$, $p < 0.05$, 95% CI [0.030–0.289]) and CY ($\beta = 0.168$, $p < 0.05$, 95% CI [0.040–0.295]) in mediation models 1 and 3, and gender did not significantly predict EX ($\beta = 0.075$, $p > 0.05$, 95% CI [$-0.047$ to 0.197]) and PE ($\beta = -0.112$, $p > 0.05$, 95% CI [$-0.255$ to 0.031]) in mediation models 2 and 4.

Furthermore, the bootstrap estimates in Table 4 revealed that the indirect effects of PSU were statistically significant in mediation models 1, 2, and 3, and PA can negatively predict burnout, EX, and CY through PSU.

## DISCUSSION

This study investigated the association between PA, PSU, and burnout with a cross-sectional survey to examine the mediating role of PSU. The findings confirmed that PSU is not only an essential contributor to burnout, but also a mechanism that mediates the impacts of PA, which will help researchers understand how to apply PA to cope with burnout.

**Table 4  Results of bootstrap estimation of indirect effects of PSU.**

|  | Effect | SE | LLCI | ULCI |
|---|---|---|---|---|
| Mediation model 1: PA→PSU→Burnout | −0.068 | 0.017 | −0.103 | −0.037 |
| Mediation model 2: PA→PSU→EX | −0.085 | 0.021 | −0.129 | −0.045 |
| Mediation model 3: PA→PSU→CY | −0.072 | 0.018 | −0.108 | −0.038 |
| Mediation model 4: PA→PSU→PE | 0.004 | 0.007 | −0.010 | 0.020 |

**Notes:**
Level of confidence for all confidence intervals in output: 95%.
Number of bootstrap samples for percentile bootstrap confidence intervals: 5,000.

## Physical activity and burnout

Self-reported data from Chinese college students illustrated that PA in the last 7 days was significantly uncorrelated with burnout, so Hypothesis 1 was not supported. That is, an enhanced level of PA does not directly lead to a reduction in burnout, and in order to prevent individual burnout, we cannot focus solely on how to achieve higher "activity levels." Of course, this is not in conflict with the fact that lack of PA is a potential threat factor for burnout (*Cheung & Li, 2019*; *Naczenski et al., 2017*). Students may need to find a specific amount of activity that "feels good" rather than thinking "more is better" (*Ladwig, Hartman & Ekkekakis, 2017*). The reason for this is that vigorous physical activity relies on intentional behavior and requires hard work that also consumes cognitive resources, while high intensity can be perceived as aversive (*Phillips & Mullan, 2022*). Intensity above the ventilatory threshold decreases pleasure, produces a bad affective response (*Ekkekakis, Hall & Petruzzello, 2008*), and to some extent hinders students from self-regulation after burnout has occurred. Nevertheless, one study showed that physical activity negatively predicted academic burnout, with the high physical activity group having lower levels of academic burnout than the moderate and low physical activity groups (*Chen et al., 2022*). We believed that the difference in results might be due to the differing measurement instruments. The Learning Burnout Scale for College Students was similarly not identical to the sub-dimensions of the MBI-SS. Therefore, to ensure that the interventions are effective emphasize differences between specific physical activities and whether the intervention outcome is a change in the exogenous or endogenous component of burnout.

## Problematic smartphone use and burnout

The results of the study showed that there was a significant positive correlation between PSU and burnout, which was consistent with Hypothesis 2. As previously stated, students with serious PSU tend to fall behind their peers in academic achievement (*Hawi & Samaha, 2016*), whether they use their smartphones excessively to socialize or for leisure (*Ekkekakis, Hall & Petruzzello, 2008*; *Van Deursen et al., 2015*). A decline in academic performance is a form of adversity for students, and the stress that comes with it means that making progress requires greater work engagement than before. Therefore, vigor, devotion, and the ability to derive new meaning and inspiration from work are essential to reducing burnout levels (*Schaufeli et al., 2002*). However, uncontrolled gaming, social media use, and short video binges have been associated with attention deficit hyperactivity

disorder (ADHD) (*Pluhar et al., 2019*), which can be a distraction from academics (*Feng et al., 2019*). Students who are not interested in the course are more likely to be attracted to the content offered by smartphones, leading to a decreased enthusiasm and motivation for learning (*Wang, Qiao & Wang, 2023*). Overall, reasonable control of students smartphone usage in terms of duration and frequency should be positive in reducing school-related burnout.

## Physical activity and problematic smartphone use

Our results demonstrated that there was a significant negative correlation between PA and PSU, and Hypothesis 3 was supported. A study by *Grimaldi-Puyana et al. (2020)* found that students with low levels of PA were three times more likely than others to increase their smartphone usage based on a subjective measuring instrument. Less PA was noted to be strongly associated with prolonged screen time and sedentary behavior, specifically talking through phone calls, texting, instant messaging, and video services (*Leatherdale, 2010*; *Rasmussen et al., 2020*). On the contrary, increased physical activity can be an effective intervention, and some randomized controlled trials have shown that exercise such as running, basketball, Baduanjin, and Qigong can reduce addiction levels in smartphone users, for instance in studies conducted by *Xiao et al. (2021)* and *Lu et al. (2020)*. The primary benefit of exercise is that it allows individuals to detach themselves from the online environment and engage in dynamic activities in the real world, shifting the focus of their attention and facilitating face-to-face social interactions with others. Furthermore, exercise can enhance self-control as a way to reduce dependence on smartphones (*Zhong, Wang & Zhang, 2021*; *Zeng et al., 2022*; *Zhang et al., 2022*), while contributing to the treatment of a range of physical and psychological symptoms caused by excessive smartphone use. Unfortunately, the current literature gap revealed that we know little about the impact of PA on PSU in other domains, except for physical exercise in leisure time, which can be a good entry point for intervention. For example, changes in the form of transport-related physical activity may also largely reduce the use of smartphones, with a tendency to choose cycling and walking instead of taking the subway or private car when travelling to school.

## Mediating role of problematic smartphone use

For the first time, we discovered that PSU entirely mediated the relationship between PA and burnout and that PSU also mediated the effects of PA on exhaustion and cynicism, with Hypothesis 4 partially supported. In real-world 24-h movement behavior, PA and PSU maintain moment-to-moment competition in time allocation, which does not include PA interventions based on smartphone-related technology support. If there is more time invested in PA, less time will be taken up by smartphones (*Fan et al., 2021*). With this, the negative consequences of excessive smartphone use are reduced, helping to further alleviate burnout syndrome.

In the physiological operating mechanism, the dorsolateral prefrontal cortex (DLPFC) is important in interfering with control tasks in brain regions (*Nigg, 2000*) that are activated in stressful situations, but acute psychological stress can cause structural and

functional changes in the DLPFC that affect cognitive inhibition in individuals. However, regular exercise can promote inhibitory control and counteract the harmful effects of stress on specific areas of the brain (*Mücke et al., 2020*). For students with excessive use of smartphones, acute aerobic exercise can also be effective in inducing changes in response inhibition and reducing PSU (*Fan et al., 2021*). We believe that this is the coping mechanism of PA for exhaustion through PSU, as stress is the main source of exhaustion.

Similarly, the results of the present study showed that cynicism is influenced by PSU and PA. For students, cynicism manifests itself specifically as a loss of interest and doubt towards their academic pursuits. As described by *Yang & Tung (2007)*, using the entertainment features of smartphones produces a pleasant experience, which in turn can contribute to an increase in behavioral frequency and deprive students of the desire to engage in learning. We were inspired by several studies around institutional employees, where researchers found a positive association between employees' participation in activity and job satisfaction (*Hutchinson & Wilson, 2012*; *Van Berkel et al., 2013*). It is appropriate to state that emotional and physical states, such as energy, inspiration, enthusiasm, dedication, struggle, and the desire to put effort, which can be regarded as reflecting the nature of sports, illustrate that students fully acclimate to these positive features.

Previous studies have shown that cardiovascular, resistance, and relaxation exercises can improve professional efficacy (*Bretland & Thorsteinsson, 2015*; *Van Rhenen et al., 2005*), but the findings of the present study contradict them, and PA did not significantly predict professional efficacy. The difference arises due to the highly purposeful nature of the complex, multistructured longitudinal intervention for company employees, where the effect of the intervention directly affects the magnitude of their productivity. However, students' self-reported PA levels cover a wide range, with only a small proportion affecting on course-related efficacy. This may mean that the "spillover" phenomenon of self-efficacy generated by PA has boundaries. Our study found that PSU did not significantly predict professional efficacy, which is different from previous related studies.

According to *Zhu et al. (2011)* and *Li, Gao & Xu (2020)*, PSU negatively predicted academic self-efficacy, and students who were addicted to smartphones were likely to be less academically competent. The explanation given was that uncontrolled smartphone use can lead to a tendency to procrastinate engaging in academic activities, which can ultimately affect learning-related confidence. However, our results contradict this, and PSU was only a important contributor to exhaustion and cynicism. As reported by *Brubaker & Beverly (2020)* there is no correlation between PSU and professional efficacy.

## Gender difference

In addition, male students were found to score higher in PA and burnout compared to female students. Most studies have confirmed that males are more active than females (*Pearson et al., 2009*; *Fan et al., 2019*), and the gender differences are mainly related to the physical education activities offered in schools, with males preferring high-intensity forms of sports (*Vašíčková et al., 2013*). In fact, male students often show more lack of interest in learning and engage in negative behaviors such as skipping class, not listening to lectures, being late, leaving early, and not turning in assignments, while female students are the

majority of the class with good grades and better overall performance. This is the reason why male students are more susceptible to burnout than female students, which is consistent with the latest survey results (*Liu et al., 2023*). However, previous studies have shown that the issue of gender differences in burnout is still highly variable (*Herrmann, Koeppen & Kessels, 2019*; *Worly et al., 2019*; *Walburg, Mialhes & Moncla, 2016*). Perhaps in the future, consistency tests for groups in cross-cultural contexts could be conducted to balance out some of the potential confounding factors.

The current study found that gender was a significant predictor of cynicism but not of exhaustion or professional efficacy. Previous studies showed that male students are more likely to exhibit cynicism and have lower professional efficacy scores, while female students report higher emotional exhaustion scores (*Worly et al., 2019*). A possible explanation for the consistency of male students' cynicism across cultures is that they tend to be more prominent in the classroom for their lack of concentration and lack of seriousness about learning. In terms of emotional exhaustion, it is a long-standing stereotype in traditional Chinese culture that women should suppress their emotions (*Hu & Schaufeli, 2009*). Gender inequality does not appear to be supported by current findings in the school setting. Female students have lower self-efficacy than male students in the learning process and are more likely to be academically compliant (*Xiao & Song, 2022*). These conflicts remain to be verified by transferring to a larger heterogeneous sample.

## Limitations and prospects

The fact that the assessments of PA, PSU, and burnout were based on self-report is a limitation of this study. Recall and social-desirability bias may impact the collection of relevant data. The cross-sectional design of the current study did not provide strong corroboration for inferences of causality, and there may be potential bidirectional effects between variables. In their investigation, *Hao et al. (2022)* noted that academic burnout leads to more process smartphone use and indirectly generates more PSU. *Unver & Buke (2022)* found that PSU negatively impacts PA levels and exacerbates musculoskeletal pain. Such results all indicate that future experimental studies with stronger evidence are needed to test the intrinsic relationships between variables.

## CONCLUSIONS

In conclusion, by investigating the association between PA, PSU, and burnout, the current study supports the perspective that PA can indirectly predict burnout through PSU. This finding provides some valuable references for developing interventions to combat burnout syndrome. To decrease possible risks to college students' health and daily routines, rationally promoting engagement with PA and cutting back on smartphone use are helpful strategies. There is no direct correlation between higher levels of PA and a lower incidence of burnout. This may imply that we should not only focus on how to achieve higher activity levels in mitigating burnout. Blindly pursuing interventions on how to enable higher levels of PA would defeat the original purpose. PA indirectly influences burnout through PSU, but does not fully apply to the three different dimensions of exhaustion, cynicism, and professional efficacy. This is an interesting finding from the present study and has inspired

more detailed insights. Only two targets of burnout (*i.e.*, EX and CY) are impacted by the indirect effects of PA through PSU. From a practical standpoint, it may be difficult for interventions through PA and PSU to be efficacious if students report burnout scores that are only worse on the professional efficacy dimension.

## ACKNOWLEDGEMENTS

We sincerely appreciate that this team chose to persevere rather than give up in the most difficult time. Meanwhile, the manuscript could not have been completed without the help of Prof. Yanmin Tan.

### Funding

This study was supported by the Special Project for Science and Technology Achievement Transformation of Qinghai Province (2023-SF-116). The funders had no role in study design, data collection and analysis, decision to publish, or preparation of the manuscript.

### Grant Disclosures

The following grant information was disclosed by the authors:
Science and Technology Achievement Transformation of Qinghai Province: 2023-SF-116.

### Competing Interests

The authors declare that they have no competing interests.

### Author Contributions

- Lianghao Zhu conceived and designed the experiments, performed the experiments, analyzed the data, authored or reviewed drafts of the article, and approved the final draft.
- Junli Hou conceived and designed the experiments, performed the experiments, authored or reviewed drafts of the article, and approved the final draft.
- Bojun Zhou analyzed the data, authored or reviewed drafts of the article, and approved the final draft.
- Xi Xiao analyzed the data, prepared figures and/or tables, and approved the final draft.
- Jingqiang Wang analyzed the data, prepared figures and/or tables, and approved the final draft.
- Wanping Jia conceived and designed the experiments, performed the experiments, authored or reviewed drafts of the article, and approved the final draft.

### Human Ethics

The following information was supplied relating to ethical approvals (*i.e.*, approving body and any reference numbers):

The study was approved by the Ethics Committee of Hubei University of Medicine (NO: 2022-TH-068).

## Data Availability

The raw data are available in the Supplemental Files.

## Supplemental Information

Supplemental information for this article can be found online at http://dx.doi.org/10.7717/peerj.16270#supplemental-information.

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
