# Peer review of "Physical activity, problematic smartphone use, and burnout among Chinese college students"

_PeerJ, doi:10.7717/peerj.16270_

## Round 0.1 · original submission · Major Revisions

I have now received the reviewers' comments on your manuscript. They have suggested some major revisions to your manuscript. Therefore, I invite you to respond to the reviewers' comments and revise your manuscript.
In addition, I suggest using newer articles in the section related to the association between physical activity and burnout.

·

Basic reporting

I would like to thank the authors for the interesting topic with rigorous review and clear aims.

For this section I have no comment.

Experimental design

The methodology was written very precise and informative but there is some points that need to be addressed by the authors to make the research more precise and sound as follows:

1- Demographic data, there is no need to write the coding of the gender, just mention that gender was included in the demographic data.

2- The cut off point score for all three scales that have been used in the study must be mentioned.

2- The reference cited for Maslach burnout inventory isnot the primary reference so please mention the original reference.

3- Authors can't use Maslach burnout inventory-student survey (MBI-SS) without permission from "Mind Garden" , so the authors has to contact them and pay for the usage of the tool and have the ethical permission and without it this tool can't be published without permission copyright holder.
The limitation that was mentioned in the supplementary file regarding this issue isn't acceptable at all.

4- Same regarding the permission to use the Smartphone addiction scale-short version (SAS-SV). Authors must pay for these tools to have permission for usage.

5- Replace coefficient alpha with "Cronbach alpha" for internal consistency.

6- The sampling design for collecting the students was missing. Was it based on probability or non probability sampling techniques?

Validity of the findings

No comment

Additional comments

The study limitations at first were clearly listed but starting from line 426, I suggest that all points being mentioned after that are not limitations and must be written in the discussion as conflicting studies and findings with the current study results.

Reviewer 2 ·

Basic reporting

See my comments.

Experimental design

See my comments.

Validity of the findings

See my comments.

Additional comments

Thanks for opportunity to review manuscript entitled ‘‘Physical activity, problematic smartphone use, and burnout among Chinese college students’’ for Peerj Journal. The strength of the manuscript includes examining variables of interest in a country where such studies are scarce. Overall, although the article is generally well written and deserves to be published in this journal some necessary and minor revisions must be made before the publication of the article. Because my main philosophy of reviewing a manuscript as reviewer and sometimes an editor to improve the manuscript and not punishing the authors, I provided very specific and detailed peer review of the manuscript to increase its quality and citation potential. I hope authors of the manuscript may benefit from my review. Necessary revisions reported section by section with the page and line number and when possible with suggestions.
1. Authors must report in the tables and in the figures with two or three decimal. Hey sometimes used two and sometimes used three decimal. Apa recommend two decimals.
2. All statistical symbols representing test statististics must be italic. For example, authors must report all r (representing Pearson correlation coefficient) mut be italic.
3. In the introduction section authors must give information about importance of their study in their cultural context. Specifically, authors need to answer ‘ ‘Why it is important to examine physical activity, problematic smartphone use, and burnout among Chinese college students using a mediation model?’’ based on previous studies and their culturxal context.
4. Authors may add research question or research hypotheses to Introduction section.
5. Following sentence is difficult to understand and must be corrected ‘ ‘In ascending order, the three levels of Cohen's d were 0.2, 0.5, and 0.8. The three levels of were 0.04, 0.25, and 0.64 (Ferguson, 2009).’’ I think authors want to mean small, medium and large effect sizes.
6. Table (e.g., Table 1) in the text must not be bold.
7. Authors must report in statistical findings section exact effect size not like this Cohen's d > 0.2.
8. The image quality is Figure 1 is very low and mus reupload. I can not read it.
9. Authors must add mean and standard deviations of variables to Table 3.
10. Significance level in the table must report low to high not like this ***p < 0.001, *p < 0.05 but like this *p < 0.05, ***p < 0.001.
11. Please correct following ‘‘Level of confidence for all confidence intervals in output: 95.’’ as Level of confidence for all confidence intervals in output: 95%’’.
12. Two figure 1 exist in manuscript authors must correct numbering. Readability of second is very problematic.
13. Practical implications of manuscript is completely missing and must be added.
14. English language editing required for this article.

---

## Round 0.2 · Major Revisions

Thank you for the update. However, there are still concerns that prevent me from accepting the revised paper. Please pay attention to the reviewers' comments and respond to them carefully.

·

Basic reporting

No comment

Experimental design

Thanks alot for addressing all the remarks being mentioned in the peer review.
I would prefer authors mentioning that they got the permission from the Malsch burnout inventory copyright holder in the manuscript.

Validity of the findings

No comment

Reviewer 2 ·

Basic reporting

See my comments

Experimental design

See my comments

Validity of the findings

See my comments

Additional comments

Thanks for opportunity to rereview manuscript entitled ''Physical activity, problematic smartphone use, and burnout among Chinese college students'' for Peer j journal. Following revisions must be made before publication of article.
1. Page 6, Line 62: Authors must provide long name of CNNIC in its first use.
2. Research hypotheses, general: I corrected one of the corrected hypothesis in APA 7 style with better scientific writing: Hypothesis 2: There will be a positive correlation between PSU and burnout. Please correct others as above.
3. Authors must add research design section to Materials & Methods. This section still completely missing.
4. Participants and procedure section must be separate. Although I indicated this in my first review, authors did not do nothing to correct it. All related information must move related sections.
5. Page 10, Line 228-229: Add age after the mean for following sentence '' with a mean of 18.55 and a standard deviation of 0.83.''
6. cronbach alpha is a special name and first letter must be as follows Cronbach alpha
7. Eta-squared classification is wrong in Line 288 289 and must be corrected. Please check https://en.wikiversity.org/wiki/Eta-squared.
8. Annual family income intersect in Table 1 and must be corrected. RMB 10,000 to 150,000 RMB 150,001 to 300,000 RMB 300,001 or more
9. M and SD must be italic in Table 1.
10. Last figure 1 in Page 34 must be deleted. it uploaded two times.

---

## Round 0.3 · accepted · Accept

In my opinion, this manuscript has been revised with attention to the reviewers' comments and can now be published.

Reviewer 2 ·

Basic reporting

Clear and unambiguous, professional English used throughout.

Experimental design

Research question well defined, relevant & meaningful. It is stated how research fills an identified knowledge gap.

Validity of the findings

All underlying data have been provided; they are robust, statistically sound, & controlled

Additional comments

Thanks for opportunity review revised manuscript entitled ‘‘Physical activity, problematic smartphone use, and burnout among Chinese college students’’. I would like the thanks to authors. They make a good job for improving quality of their manuscript. Authors revised the manuscript as I requested with a good will. In this form, Introduction reflects very well the previous studies and study aim, Method section and Result section is correct, and Discussion section adequately synthesis to previous study findings and current study results. Overall, I have no further comment regarding to manuscript. I congratulate to authors and wish them success on their future endeavors.